# Multi-Frequency Resonance Behaviour of a Si Fractal NEMS Resonator

**DOI:** 10.3390/nano10040811

**Published:** 2020-04-23

**Authors:** Vassil Tzanov, Jordi Llobet, Francesc Torres, Francesc Perez-Murano, Nuria Barniol

**Affiliations:** 1Department of Electronics Engineering, Engineering School, Universitat Autonoma de Barcelona (UAB), 08193 Bellaterra, Spain; Francesc.Torres@uab.cat (F.T.); nuria.barniol@uab.cat (N.B.); 2International Iberian Nanotechnology Laboratory (INL), 4715-330 Braga, Portugal; jordi.llobet@inl.int; 3Institut de Microelectronica de Barcelona (IMB-CNM CSIC), 08193 Bellaterra, Spain; francesc.perez@imb-cnm.csic.es

**Keywords:** NEMS, fractal structure, self-similarity, nanoresonator, broadband frequency spectrum, nonlinearity, piezoresistivity

## Abstract

Novel Si-based nanosize mechanical resonator has been top-down fabricated. The shape of the resonating body has been numerically derived and consists of seven star-polygons that form a fractal structure. The actual resonator is defined by focused ion-beam implantation on a SOI wafer where its 18 vertices are clamped to nanopillars. The structure is suspended over a 10 μm trench and has width of 12 μm. Its thickness of 0.040 μm is defined by the fabrication process and prescribes Young’s modulus of 76 GPa which is significantly lower than the value of the bulk material. The resonator is excited by the bottom Si-layer and the interferometric characterisation confirms broadband frequency response with quality factors of over 800 for several peaks between 2 MHz and 16 MHz. COMSOL FEM software has been used to vary material properties and residual stress in order to fit the eigenfrequencies of the model with the resonance peaks detected experimentally. Further use of the model shows how the symmetry of the device affects the frequency spectrum. Also, by using the FEM model, the possibility for an electrical read out of the device was tested. The experimental measurements and simulations proved that the device can resonate at many different excitation frequencies allowing multiple operational bands. The size, and the power needed for actuation are comparable with the ones of single beam resonator while the fractal structure allows much larger area for functionalisation.

## 1. Introduction

Top down nano-electromechanical structures used as NEMS resonators are extensively investigated due to their application in sensing devices. They have small mass, high quality factor and resonance frequencies. Also nanoresonators ensure precision and ultra-fast response when used as mass sensors or AFM probes [1,2,3]. The material dimensions of the nanostructures are extremely small and they exhibit material properties of dissimilar values compared to their bulk analogues [4]. Moreover, the importance of crystalline orientation can not be neglected [5]. Thus, detectable changes of mechanical and electromagnetic properties are also employed for detection [3,6].

Due to its simple geometric shape and approximate 1D-dimension, nanobeam is a preferable choice for a resonating structure. Nanobeams became functionalised for wide variety of mass detectors of extremely small objects of interest in Physics, Chemistry and Biology [1,7,8,9,10]. Moreover, there is sufficiently good prediction of nanobeams’ resonant frequency and modeshapes by using beam models or finite element method (FEM) softwares such as ANSYS or COMSOL [11,12]. That is why nanobeam resonators are the most developed so far and most of the materials suitable for nanoresonator fabrication had already been tested as a nanobeam resonator [13,14]. A drawback of beam geometry is that coupled mode vibrations easily occur. Then, the individual harmonics that produce the overall oscillation are hard to be differentiated [15,16]. The problem can be resolved by an array of nanobeams where each beam is responsible for a particular frequency [17,18]. This way, each nanobeam is responsible for a particular frequency of interest and coupled modes can be avoided. Due to relatively large circuit elements, most often, these solutions does not compromise the overall size of the devices. However, for the sake of miniaturisation, the size of the multifrequency devices must be further optimised. Twenty years ago, radio frequency devices get miniaturised by the fractal shaped antennas. That opened a scientific field that is still not fully explored. By designing fractal shaped antennas engineers are able to have the functionality of an array of dipole antennas while occupying much smaller volume [19,20]. Also, during the last ten years, fractal designs has been utilised in many novel micro- and nano-devices. More specifically, fractal micromixers have been developed to rapidly stir nano- and micro-liter volume solutions [21,22]. In photonics, fractal shapes are widely investigated where surface plasmon resonance is observed [23,24]. In the field of MEMS RF capacitors, fractal geometries are studied for suppressing residual stress, miniaturisation, excitation by single layer capacitance and optimisation of pull-in instability effects [25,26,27].

Regarding NEMS, to the knowledge of the authors, fractal shape has not been yet utilised as an electrostatically driven mechanical resonator. Hence, the current work is a pioneer investigation of the functionality and behaviour of oscillating mass that has self-similar geometry of nanoscale dimension. Our device consists of seven identical 62 star polygons generated by an Iterated Function System (IFS) that forms Sierpinsky flake fractal of second iteration [28]. In our work we show that as in the case of large scale fractal structures [29] the fractal geometry allows multiple resonant frequencies. The quality factor of the resonant peaks is comparable with the one measured in nanobeams fabricated by the same technique. This result can be associated with several features that are present in our device. Firstly, the truss geometry and superior span-to-volume ratio of fractals that allows the resonating body to span relatively wide area for the given mass while still being a rigid structure. A different truss device with similar properties was also exploited in the work of Heritier [2] where two nanowires have been connected by nanostruds forming a nanoladder for AFM probe application. They report sensitivity of a nanowire detector combined with rigidity of a cantilever beam. Secondly, as our fractal structure is clamped at all its vertices and the fabrication technology introduces high residual stress, we expect to some degree the effects of dissipation dilution and higher Q due to clamp-tapering [30,31]. In the recent studies of 1D and hexagonal phonon crystals (PnC) [32,33,34] these enhancing Q effects are combined with soft clamping which further reveals the importance of the lattice geometry for the amount of dissipated mechanical energy through the anchors.

In our work we report fabrication process that allows nanoscale functional Silicon structures of complicated fractal shapes. Our experimental setup consist of electrostatic excitation in vacuum and an interferometric system that provides optical readout. By variation of the AC-voltage, using lock-in amplifier we systematically explore different resonance phenomena. We experimentally detected multiple resonance peaks in the range of [2 MHz, 24 MHz] where linear and nonlinear frequency response behaviour was examined. We report quality factors and low excitation energy that are comparable with nanobeams fabricated by the same technique. The FEM simulations were done by COMSOL allowing for stationary study to define the residual stress. Then, the resonance modes are explored by using eigenvalue together with frequency domain studies. We have used COMSOL to show how the symmetry and stress of the resonator affect the resonant modes. Moreover, we use COMSOL to show that the resonant peaks can be detected not only by optical but also, by electrical measurement setup.

The main advantage of the structure is its self-similar geometry. Our fractal resonator anchored with nanopillars has the ability to resonate at many different frequencies while keeping quality factors high enough for detection purposes. The specific shape also ensures very compact design and high span-to-volume ratio. We want to emphasise our versatile fabrication process that allows functional Si-based devices with complex design ready for investigation. Therefore, we have shown that using this top-down technique one can experiment with sophisticate geometries.

The work is structured in five main sections and supporting material is given. In Section 2, the details of the fabrication process and design characteristics are given. In Section 3, the measurement setup is described followed by the experimental results. In Section 4, the simulation results are presented and alternative approaches are discussed. In Section 5, complementary and future work is discussed as well as side results that are included in the supplementary material.

## 2. Device and Fabrication

The present work is a proof of concept of multifrequency mechanical resonator at nanoscale whose top and side view SEM images are shown in Figure 1a,b. To this end we use the top-down fabrication process based on direct focused ion beam (FIB) implantation which was previously employed for nanobeams [35]. The resonating body of the device consists of seven star-polygons that form a fractal structure. Its all 18 vertices are clamped to nanopillars that are 2 μm long and connect the resonator with the insulating SiO_2_ layer. The gap between the hanged structure and the insulator is produced when the 2 μm thick top Si-layer has been etched. The middle SiO_2_-layer has thickness of 2 μm too. The structure is suspended over a 10 μm trench and has width of 12 μm. The nanobeams that form the device are 0.12 μm wide and 0.04 μm thick. The thickness of 0.04 μm is defined by the fabrication process and prescribes Young’s modulus of 76 GPa which is significantly lower than the value of the bulk material.

For the fabrication of the devices we diced 1 × 1 cm^2^ chips from a silicon on insulator wafer with 2 μm thick device layer oriented in the 100 crystallographic direction. Before the fabrication, each dice have been cleaned in an organic solvent (acetone) to remove potential organic contamination and rinsed in isopropanol. The samples have been directly patterned by focused ion beam (FIB) implantation of Ga+ at 30 keV, beam current of 10 pA and dose of $1\times 10^{16}$ per cm^2^ in a Cross-Beam system (1560xB from Zeiss, Oberkochen, Germany). The ion beam has been controlled with the nanolithography kit Elphy Quantum (from Raith, Dortmund, Germany) to enable the definition of different geometries [11]. It is remarkable to mention that this resist free lithographic step permits to modify the crystallinity of the irradiated silicon creating an etching mask that can be functional from the electrical point of view with the appropriate treatment [35]. According to SRIM (The Stopping and Range of Ions in Matter) simulations, the implanted and damaged volume in silicon goes up to 40nm in depth when ions goes perpendicular to the surface. This has been concluded from the TEM measurements after the ion implantation [35] and the SEM images after the silicon etching [36]. Working at the above mentioned conditions the thickness of the devices is fixed at 40 nm. Supporting pillars have been defined by ion milling and lateral ion implantation to sustain the fractal resonator, in a similar way as in [37] where a suspended lateral electrode was developed for the gatebility of suspended single charged transistors. Tetramethylammonium hydroxide at 80 degree Celsius and 25% concentration has been prepared to selectively etch the non-implanted volume of silicon. This approach permits to fabricate free suspended mechanical structures in only two steps [38]. The fractal pattern has been generated by an equation based home-written Python code which is able to produce wide variety of polygonal fractals while ensuring that the substructures of the resulted fractal shape do not overlap; see [28]. We decided for a structure that consist of 62-polygons where the self-similarity is up to second iteration [39] in the sense of IFS. Finally, in order to improve the electrical conductivity, we put through the devices under an annealing process at high temperature (up to 1000 °C) with boron in a nitrogen reach atmotmosphere to promote the recrystallization and doping (p-type) of the device [35,38]. This final step serves two important purposes for the electrical excitation of the devices to be achieved. It promotes the crystallisation of the amorphous material and it introduces electrical carriers through their doping.

## 3. Experimental Set-Up and Characterisation

The devices have been measured in an in-house heterodyne interferometric set-up which allows the optical characterisation of the resonant displacement of NEMS devices in air and in vacuum environment. This system is composed by a vacuum chamber to place the sample, a 633 nm He-Ne laser, an acousto-optic modulator with 40 MHz of frequency shift, an avalanche photodetector of 1 GHz bandwidth which is connected to a 50 MHz Zurich Instrument lock-in amplifier that performs the read-out of the interference signal produced at the photodetector. Our interferometer is placed over an optical table with active vibration isolation. The estimated maximum resolution for the out-of-plane amplitudes measurement is at about 4 nm.

The layout of the device includes two pads where the electrical current can be feed-in while the bottom Si-layer of the chip has been grounded, [40]. Using these pads, IV-curve measurements of nano-crystalline doped silicon (nc-Si) and amorphous silicon (a-Si) structures were conducted. First, an a-Si structure was measured showing an ohmic behaviour with resistance of 33.33 MΩ. Then, an nc-Si structure with improved conductivity was also measured exhibiting an ohmic behaviour with much lower resistance of 420 kΩ. Taking the geometry of the structure into account the resistivity of the annealed structure was computed by COMSOL to be 0.0024 Ω/m. Consequently, the a-Si structure was not able to be excited due to the poor electrical conductivity while the electrically improved nc-Si structures were tested to be functional. The performed measurements use one of those pads to apply alternating current $V_{AC}$ producing potential difference with the ground in order to excite the structure electrostatically leading to transverse mechanical vibration. The improvement of the excitation due to doping in the Si-nanodevices shows that the annealing step reduces the resistivity of the material [41] allowing higher voltage to be distributed further away from the excitation pad of the device. Thus, the electrostatic force between the suspended structure and the ground layer becomes large enough for the resonator to function.

By using the described set-up, the experimental measurements confirm multiple resonance peaks from 2 MHz up to 24 MHz; see Figure 2. The laser spot of about 12 μm just covers the whole structure and we have aligned its centre with the centre of the structure in order to equally cover the membrane. The usual measurement condition we have used between 2 MHz and 17 MHz was peak-to-peak voltage V${}_{pp}$ = 18 V and offset voltage V${}_{off}$ = 0. However, the magnitude of the response is very sensible to the experimental conditions and ranges above 18 MHz need additional measurements with V${}_{off}$ = 3 V for the targeted resonant peaks to become clear among the others. The resulted frequency response spectrum has multiple nonlinear resonance peaks where the peaks up to 15 MHz are presented in detail. When measurements at the specific peaks were conducted the softening (negative amplitude dependent frequency shift) and hardening (positive amplitude dependent frequency shift) were detected, hence they were swiped-up and -down for different voltages so the hysteresis jumps to become evident. We have also detected the motion of the structure in air by applying up to 40 V DC bias voltage, see Section A.2.

### 3.1. The Peaks at 2.37 MHz and 5.1 MHz

In the frequency range from 2.3 to 5.2 MHz we measured linear and nonlinear responses. The peak at 2.37 MHz, see Figure 3a, is an example for nonlinear hardening, where the hysteresis loop becomes larger when V${}_{off}$ is applied. For lower excitation rates, linear behaviour was detected and Q factor of 805 was calculated. All measured Q-factors can be seen in Section A.3. Interestingly, the peak at 5.1 MHz shown in Figure 3b decreases when V${}_{off}$ is applied. The peak at 5.1 MHz preserves its linear behaviour for all excitation voltages we tried while spanning the allowed range of our equipment. The Q factor for $V_{pp}$ = 20 V, V${}_{off}$ = 0 V was estimated to be Q ≈ 912 computed at −3 dB [3]. It should be mentioned that other peaks at higher frequencies also decrease its maximum amplitude when the offset voltage V${}_{off}$ is applied; see Section A.1.

### 3.2. The Peaks at 6.1 MHz and 7.1 MHz

The peak at 6.1 MHz in Figure 4a is an example for nonlinear softening combined with hardening, where there are two hysteresis loops that form an inclined plateau. For lower V${}_{pp}$ the peak does not have the hysteresis loop from the right-hand side and looks like nonlinear softening. It has been used as a control peak during the measurements. According to the simulations, this peak corresponds to the odd-(1,1) ellipse membrane mode; see Section 4.1.

The double peak at 7.1 MHz shown in Figure 4b appears from a single peak when V${}_{pp}$ is increased up to V${}_{pp}$ = 18 V. Also, when V${}_{off}$ increases it further splits the peak making the hysteresis loops more evident. Few other peaks have similar behaviour, where increased V${}_{off}$ divides a plateau-like peak. Barely visible hysteresis loops were observed, hence they are not plotted for clarity.

### 3.3. The Peaks at 9.8 MHz and 10.1 MHz

Between 9.2 MHz and 10 MHz there are three main peaks. In panel Figure 5a the ones at about 9.8 MHz are shown. If V${}_{off}$ is applied, another peak at 9.6 MHz merges with the ones shown here. For V${}_{pp}=10$ there are two peaks (see the blue curve), one just below 9.8 MHz and another one at 9.83 MHz. When V${}_{pp}$ increases to 18V, the double inclined plateau and a hardening hysteresis loop become visible; see the cyan and red curves in Figure 5a.

A resonance peak appears at 10.1 MHz (Figure 5b) which is twice the frequency of the peak shown in Figure 3b. When the peak-to-peak voltage is increased a hardening reveals; see the cyan curve of Figure 5b.

There is a peak at 11.2 MHz but it was impossible to be detected in a great detail. Thus, we did not include a dedicated figure for the corresponding range of the spectrum. It does not have significant hysteresis loops but sometimes appears as a very noisy triple peak.

### 3.4. The Peaks at 12.3 MHz and 13.05 MHz

The peak at 12.3 MHz shown in Figure 6a appears at frequency twice bigger than the frequency of the control peak; see Figure 4a. Moreover, the softening type nonlinearity is preserved and clearly visible for higher V${}_{pp}$. However, no significant hysteresis loop was found, hence the swipe-down is skipped in the figure.

In Figure 6b the peak at about 13.05 MHz is shown. It exhibits interesting phenomena where the increase of V${}_{pp}$ shifts the linear peak to higher frequencies. We were not able to experience nonlinear hardening or softening in the range of voltages we operate. This way, the peak’s resonance frequency increases with the applied voltage without experiencing nonlinearity. The quality factor for $V_{pp}$ = 20 V (yellow peak) was estimated to be Q ≈ 961.

### 3.5. The Peaks at 13.54 MHz and 14.3 MHz

The peak at 13.45 MHz (see Figure 7a) exhibits notable softening type nonlinearity when V${}_{pp}$ is increased. The detected hysteresis loop spans over a range of 50 kHz.

Figure 7b shows us how at 14.3 MHz when V${}_{off}$ is increased, the double peak and the corresponding hysteresis loops become more notable; see the cyan and the red peaks in Figure 7b. This double peak appears at twice the frequency of the double peak shown in Figure 4b. Again, if only V${}_{pp}$ is increased for V${}_{off}$ = 0 V, a single peak evolves to an inclined plateau and then if further increased it divides into two separate peaks.

We have initially swiped up and down every peak under the conditions of V${}_{off}$ = 0, V${}_{pp}$ = 10 and V${}_{off}$ = 0, V${}_{pp}$ = 18. Then, we applied many different V${}_{pp}$ voltages up to V${}_{pp}$ = 20 where we reached the limit of our instrument. In that fashion, we recorded the interesting nonlinear behaviour presented in this section and in Section A.1. However, for higher frequencies above 18 MHz we needed to use V${}_{off}$ in order to obtain better response. This way, we have seen that V${}_{off}$ influences the peaks dissimilarly from V${}_{pp}$. For example, double peaks widen when V${}_{off}$ is increased and some peaks get lower when V${}_{off}$ is increased. Hence, we did more measurements of the peaks below 18 MHz for different V${}_{off}$ conditions. These experimental findings are interesting and their theoretical explanation would be beyond the scope of the present study. One approach would be if nonlinear approximations using Duffing-type equations are applied to the nonlinear peaks which might help to determine the Young modulus; see the Discussion Section 5. In the following section we find this important material property by using our FEM model.

## 4. FEM Simulations

We use COMSOL finite element method (FEM) software in order to define the fractal resonator and to investigate its eigenfrequencies and material properties. Different studies were set so we can distinguish how the geometry of the device and the residual stress affect the frequency spectrum of the device. In the following Section 4.1 we compare the resonant frequencies of two hexagonal-based fractal resonators. One, where sections AD = BE = 12 μm but CF = 10 μm with another one where the sections AD = BE = CF = 12 μm, see Figure 8. Then, in Section 4.2 an initial stress is introduced to the device. As it was previously shown with a nanobeam [11], the stress can alter the order at which the different resonant modes appear while swiping the frequency spectrum. Finally, in Section B.2 Frequency Domain study was set to show that resonant peaks affect the resistance of the device, hence they can be detected by electrical measurements.

The extremely thin resonator we are modelling needs careful investigation of how the reduced size influence the material properties. An important size dependent property that we take into account is the Young’s modulus of polysilicon which for the bulk material has to be at about 160 GPa. According to [4] a Si-film of 50 nm thickness has Young’s modulus of about 70 Gpa, which is significantly smaller compared to the one of bulk Si. In order to define the value of Young’s modulus we have modelled a double clamped nanobeam of length 4.16 μm, width 540 nm and thickness 40 nm that was fabricated by the same top-down technique explained above in Section 2. The purpose was to fit the resulted eigenvalues with the measurements shown in [11]. As it is shown from the comparison between the simulated and experimental natural frequencies of Si-nanobeam in Section 4.2, the first four resonant frequencies have very similar values. This was possible by reducing the Young’s modulus down to 76 GPa and introducing stress to the structure in order to resemble the effect of post fabrication residual stress. Thus we presume the elastic properties of polysilicon were calibrated well for our fabrication technique.

At this point we have changed the nanobeam geometry and introduced the clamped fractal structure geometry in the COMSOL toolkit. This was achieved by a vertical extrusion of previously computed 2D vector shape of the structure’s horizontal section. Then, by the Solid Mechanics Interface we defined the material properties Young’s modulus, Poisson ratio and density, and the boundary conditions. In our model we have the resonator that has fixed boundary conditions at its vertices and two block domains that correspond to the gap and the insulation SiO_2_ layer. The insulation layer has fixed boundary conditions as well. Also, we define the Electrostatic Interface where the excitation pad and the ground(the lower boundary of the SiO_2_ layer) are specified. The next step is to mesh the domains. As the height of our device is relatively small in comparison with its length and width, we first use triangle mesh for the upper boundary of the structure which has been swiped down through the domains. When the mesh was evaluated, we need to define the different solvers. For the eigenfrequency study of the relaxed structure shown in the following subsection we use only the eigenfrequency solver. For the investigation of the natural frequencies of the compressed structure that computes the results of Section 4.2, a previous study that defines the bending must be run. The bending is computed by a Stationary study where the boundary conditions that define the side electrodes are displaced towards each other. Furthermore, DC voltage is applied and then gradually decreased to 0V so the structure is firstly attracted towards the bottom Si-layer and then left at its bent equilibrium. The resulted bent shape has its specific stresses that resemble the postfabrication stress, hence this numerical output is plugged again in the Eigenfrequency study where we compute the eigenfrequencies of the compressed structure. For both, Eigenfrequency and Stationary studies we use the standard solver settings of COMSOL. The eigenvale solver uses MUMPS direct solver while the stationary study uses combination of MUMPS and segregated solver where the mechanics and electrostatics are decoupled

### 4.1. Symmetry Axes and Mode Shapes

The simulations shown in Figure 8 represent the mode shapes of the fabricated device from Figure 1. Due to the length difference between the section CF and AD = BE the resulted geometry is symmetric only about two axes. Namely, $\vec{Ox}$(FC) and $\vec{Oy}$ where *O* is the center of the fractal. However, if AD = BE = CF then these three sections will lie on three different axes of symmetry. The effect of such symmetry breaking can be seen in Table 1 where the resulted resonant frequencies are given. In both cases the first five mode shapes correspond to elliptic membrane mode shapes (see Table 1). About membrane’s modes notation, see [42]. The difference is that the odd- and even-symmetrical variations of (1,1)- and (2,1)-mode split and appear at different frequencies if AD = BE ≠ CF. Thus, if AD = BE ≠ CF the frequency spectrum has less double resonance peaks than in the case of AD = BE = CF.

### 4.2. Mode Alternation Due to Compression Stress

The fabrication process of the device consists of three steps; see Section 2. Previous study shows, that this fabrication process results in a compression stress due to the Gallium implantation that causes nanobeams to buckle [11]. By using COMSOL Eigenvalue study the compression stress was introduced and the first four natural frequencies of the nanobeam were successfully simulated. It was shown that the mode with three nodes appears before the two nodes mode. In other words, what is known as second mode shape appears at lower frequency than the first mode shape, see Table 2, columns 1, 2 and 3. The needed effect of compression was achieved numerically by displacing the side electrodes toward each other. By using the same technique, an eigenvalue study of compressed fractal structure was defined. In COMSOL, the distance between the resonator’s pads was decreased by 4 nm. That way we defined buckling with 82 nm displacement at the center of the structure. However, for the buckling to be achieved numerically some voltage must be applied. Then, the voltage was decreased to zero so the model could reach the stable equilibrium at 0 V where the structure has buckled shape. Few different simulations were run with various voltages to verify that the achieved buckled shape is independent of the applied force. Also, as the fabrication technique is the same as in the case of nanobeam, the values of Young modulus = 76 GPa, Poisson ratio = 0.3 and density = 2328 kg/m^3^ are also kept the same as in the successful nanobeam model [40]. Finally, an eigenvalue study was performed where the temperature was set to 296 K. In Table 2 we compare the frequency measurements of the first five modes with the frequencies resulted from the numerical study and the first three of them fit very well. When the natural frequencies of the device before and after compression get compared (see Table 1 and Table 2), we detected that the standard order of appearance of the elliptical membrane modes is altered due to the stress of the device. Without compression we have even-(0,1), odd-(1,1), even-(1,1), even-(2,1) and odd-(2,1), and after compression we have even-(1,1), even-(0,1), odd-(1,1), odd-(2,1) and even-(2,1) modes.

## 5. Discussion

Our resonator employs a nontrivial shape aiming to show that self-similar structures of very small size can be used as mechanical resonators. The functionality get proved and multiple resonant peaks were detected. We comment on the nonlinearities detected at most of the resonances but we do not try to model them due to uncertainties that are beyond the scope of the present work. Nevertheless, let us discuss the evident nonlinear behaviour of our system.

Experimentally, we can categorise the different types of nonlinearities that are evident. Nonlinear effects of softening and hardening, and superharmonics that get excited at twice the resonant frequency are evident. Double peaks that get divided by increased bias voltage are also evident. Our measurement set-up does not have enough resolution to detect the displacement of the individual substructures of the device. Thus, we were unable to get clear answer what caused these nonlinear effects. One option is the frequency of the elliptic-like membrane modes to shift with the amplitude. However, if there is a prominent localisation phenomena [43,44], then some resonances of the substructures can get excited and coupled modes will explain the nonlinear peaks.

Theoretically, by utilising nonlinear Duffing-type equations for modelling nanobeams and 2D-membranes, one can explain spring hardening and softening phenomena with the initial stress and the excitation voltage, see [45,46] and the references therein. Thus, by using the coefficients in front of the nonlinear terms, one can fit a nonlinear resonant curve for a particular mode and obtain some material properties such as Young’s modulus [46]. Moreover, doping and crystal orientation can influence the nonlinear response of capacitive doped Si MEMS resonators [45]. Also, superharmonic resonance is a nonlinearity induced by the second power of the harmonic term of the capacitive forcing [45]. And finally, double peaks could be due to symmetry breaking.

The fractal structure of our interest does not allow proper vibration modelling by utilising well-known membrane equations due to its specific geometry that brings the following arguments. As seen in Figure A12 of the Appendix, if our device is stretched along x-axis this would lead to elongation along y-axis too. Thus, our device can be associated to the materials with structural hierarchy (truss structures) which have negative Poisson ratio [47]. At resonance, this property will lead to strain and stress components along y-direction that are of opposite sign if compared with the ones produced by non-fractal 2D-membrane. Another factor that can affect the nonlinear response is the doping and our structure was doped once for the shape to be defined, and once again during the annealing step. As doping affects the resistivity [41] it affects the voltage distribution which variations along the structure lead to electrostatic nonlinearities. Also, doping changes the material’s mechanical properties due to induced defects in the crystal lattice, hence, can lead to nonlinear response. Indeed, doping can cause both spring hardening and softening phenomena; as shown in [45]. Taking the mentioned above into account we do not focus on modelling the nonlinear behaviour. Instead, we used the opportunity to define the geometry of the device, its material properties and physical interactions into a multiphysics FEM model. In [12,29,34,43] FEM is successfully employed for the analysis of the properties of a vibrating fractal structures and phononic crystals. Thus, we use COMSOL FEM software to vary the stress in x-direction so the resulted eigenmodes fit with the experimental ones. Moreover, in the Section B.2 the piezoresistivity of the device is simulated by our FEM model for the case of non-compressed fractal structure. This way, we show that many applications of our device that would rely on electrical resonant peak detection can be achieved.

The two most important application examples are mass and force sensing which are very convenient for NEMS devices. The broadband spectrum of the device allows different types of multi-frequency experiments to be conducted [48]. Also, the multiple double peaks controlled by $V_{off}$ (see Figure 4b, Figure 5a and Figure 7b) can be employed as bandpass filters. Arrays of nanobeams are employed for bio-applications on molecular level where high sensitivity is needed but single nanobeam can not provide enough area for a practical device [6]. Taking this into account, the fractal geometry has a beneficial property of high span-to-volume ratio. This ensures relatively large area for sensitive biological element to be placed without compromising with additional mass. If localisation effects are able to be employed, each substructure can be functionalised differently resulting in a multi-analyte bio-sensor.

## 6. Conclusions

The present work is dedicated to the fabrication, measurement and analysis of a NEMS device which uses a fractal nanostructure as a suspended resonating body that is capacitively excited. The fabrication of the device is defined by using Ga FIB-implantation, wet etching and annealing in Ba environment. It has been characterised by an interferometric set-up which allows optical characterization in air and vacuum environment. Then, we optically proved the functionality of all our suspended fractal structures as NEMS mechanical resonators. We have detected broadband spectrum where we specified the quality factors of most of the peaks. We also associated the first five peaks with the resonant modes found by our COMSOL FEM model. Then, by utilising our model, we found how the symmetry of the device can affect the resulted spectrum and how the residual stress can affect the distribution of the mechanical modes. Moreover, with COMSOL we found that due to piezoresistivity, resonant peaks can be detected by measuring the resistance of the device built between the pads.

## Figures and Tables

**Figure 1 nanomaterials-10-00811-f001:**
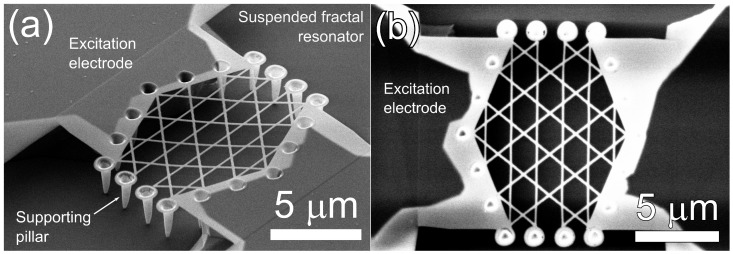
SEM images of the fractal nano structure: (**a**) 45 degree tilted SEM image of the suspended fractal resonator. (**b**) Top-view SEM image of the suspended structure.

**Figure 2 nanomaterials-10-00811-f002:**
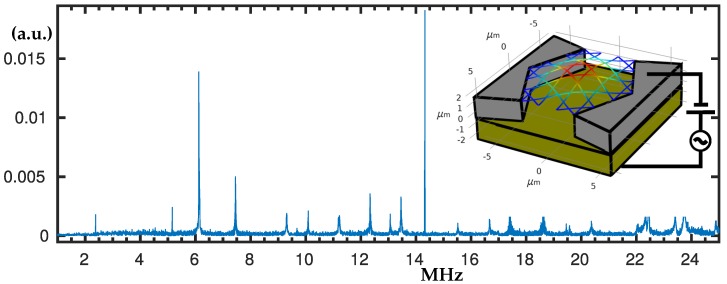
Experimentally measured frequency response spectrum produced by using electrostatic actuation and optical interferometric detection.

**Figure 3 nanomaterials-10-00811-f003:**
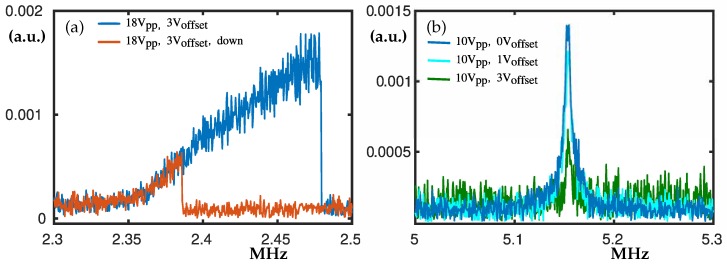
Experimental frequency swipes centred at about 2.37 MHz and 5.15 MHz. In panel (**a**) the blue curve denotes swipe-up and the red curve denotes swipe-down. The excitation AC voltage is 18 V peak-to-peak (V${}_{pp}$ = 18 V) and 3 V offset (V${}_{off}$ = 3 V). In panel (**b**) V${}_{pp}$ = 10 V where the blue, cyan and green curves denote V${}_{off}$ = 0 V, V${}_{off}$ = 1 V and V${}_{off}$ = 3 V, respectively.

**Figure 4 nanomaterials-10-00811-f004:**
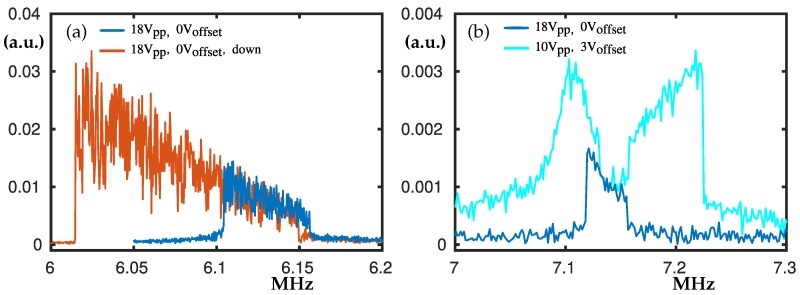
Experimental frequency swipes centred at about 6.1 MHz and 7.1 MHz. In panel (**a**), V${}_{pp}$ = 18 V, V${}_{off}$ = 0 V where, the blue curve denotes swipe-up and the red curve denotes swipe-down. In panel (**b**) the blue and cyan curves denote V${}_{pp}$ = 18 V, V${}_{off}$ = 0 V and V${}_{pp}$ = 10 V, V${}_{off}$ = 3 V, respectively.

**Figure 5 nanomaterials-10-00811-f005:**
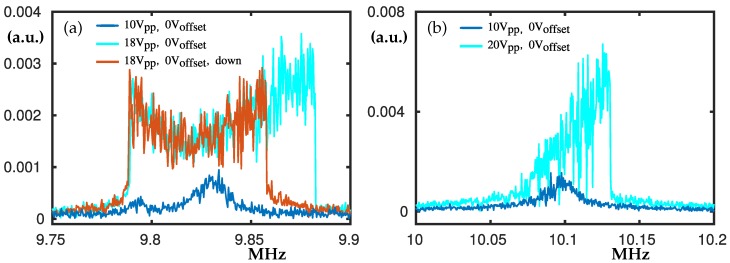
Experimental frequency swipes centred at about 9.8 MHz in panel (**a**) and 10.1 MHz in panel (**b**). In panel (**a**) V${}_{off}$ = 0, where the blue curve denote swipe up for V${}_{pp}$ = 10 V and the cyan and red curves denote swipe up and down for V${}_{pp}$ = 18 V. In panel (**b**) V${}_{off}$ = 0 V and the blue and cyan curves denote swipe-up for V${}_{pp}$ = 10 V and V${}_{pp}$ = 20 V, respectively. The peak at about 10.1 MHz appears at twice the frequency of the peak shown in Figure 3b.

**Figure 6 nanomaterials-10-00811-f006:**
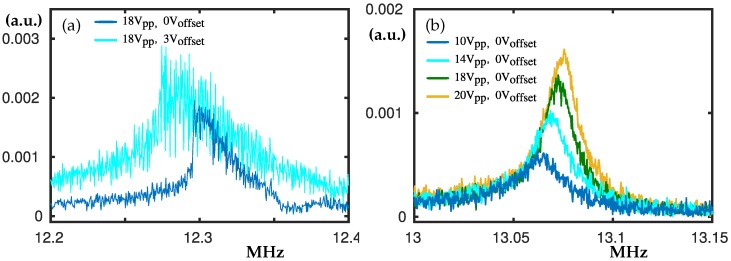
Experimental frequency swipe up centred at about 12.3 MHz and 13.05 MHz. In panel (**a**) V${}_{pp}$ = 18 V. The blue and cyan curves denote V${}_{off}$ = 0 V and V${}_{off}$ = 3 V, respectively. In panel (**b**) V${}_{off}$ = 0 V. The the blue, cyan, green and yellow curves denote: V${}_{pp}$ = 10 V, V${}_{pp}$ = 14 V, V${}_{pp}$ = 18 V, V${}_{pp}$ = 20 V. The peak at about 12.3 MHz appears at twice the frequency of the peak shown in Figure 4a.

**Figure 7 nanomaterials-10-00811-f007:**
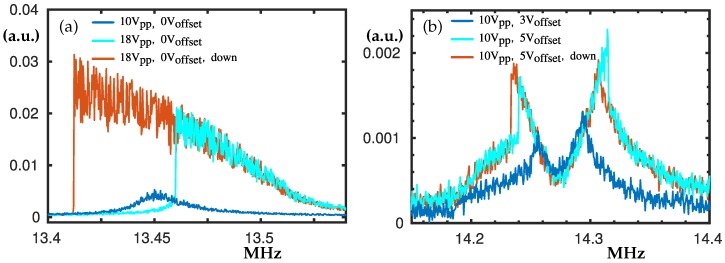
Experimental frequency swipes centred at about 13.45 MHz and 14.3 MHz. In panel (**a**) V${}_{off}$ = 0 V. The blue curve denote V${}_{pp}$ = 10 V while the cyan together with the red curve are the swipe-up and -down for V${}_{pp}$ = 18 V. In panel (**b**) the blue curve denotes swipe up for V${}_{off}$ = 3 V and V${}_{pp}$ = 10 V. Cyan and red curves denote swipe-up and -down for V${}_{off}$ = 5 V and V${}_{pp}$ = 10 V. The peak at about 14.3 MHz appears at twice the frequency of the peak shown in Figure 4b.

**Figure 8 nanomaterials-10-00811-f008:**
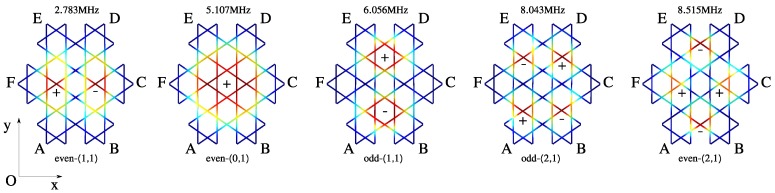
The first five resonant modes of vibration and their corresponding frequencies for the compressed fractal structure. Red colour denotes maximum displacements while + and − signs denote the positive and negative amplitudes along z-axis (orthogonal to the xy-plane). In Table 1 and Table 2 the frequencies at which the modes appear for the uncompressed and compressed structures are shown.

**Table 1 nanomaterials-10-00811-t001:** The first five resonant frequencies for the cases where AD = BE = CF amd AD = BE ≠ CF. The different mode shapes can be seen in Figure 8.

Sections	AD = BE = CF	AD = BE ≠ CF
**Compression**	0 nm	0 nm
**Maximum Bending Displacement**	0 μm	0 μm
even-(0,1)	3.176 MHz	3.837 MHz
odd-(1,1)	6.577 MHz	7.428 MHz
even-(1,1)	6.577 MHz	8.422 MHz
even-(2,1)	10.654 MHz	12.106 MHz
odd-(2,1)	10.654 MHz	13.004 MHz

**Table 2 nanomaterials-10-00811-t002:** Table of the first five resonant modes together with their frequencies for compressed beam, Columns 2 and 3, and the compressed fractal structure, Columns 5 and 6. The last row shows the fitted material properties of the suspended structures.

Nanobeam			Fractal resonator		
**Compression**		8.6 nm	**Compression**		4 nm
**Maximum Bending Displacement**		114.6 nm	**Maximum Bending Displacement**		82 nm
**Temperature**		298 K	**Temperature**		296 K
**Mode**	Experimental	Numerical	**Mode**	Experimental	Numerical
second	26.2 MHz	25.97 MHz	even-(1,1)	2.4 MHz	2.783 MHz
first	30.9 MHz	31.255 MHz	even-(0,1)	5.15 MHz	5.107 MHz
torsional	71.8 MHz	71.826 MHz	odd-(1,1)	6.1 MHz	6.06 MHz
third	76.9 MHz	76.322 MHz	odd-(2,1)	7.15 MHz	8.043 MHz
fourth	96 MHz	106.48 MHz	even-(2,1)	9.22 MHz	8.5151 MHz
**Material Properties**		Young modulus = 76 GPa; Poisson ratio = 0.3; ρ = 2328 kg/m^3^			

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
