# Peer review of "Multi-Frequency Resonance Behaviour of a Si Fractal NEMS Resonator"

_nanomaterials, 2020, doi:10.3390/nano10040811_

Round 1

Reviewer 1 Report

The authors present a nanoresonator that is interesting and original for two main reasons:

  1. The resonator is produced by an interesting, maskless method that I have not seen often before.
  2. The factal structure is of interest, in my view mainly because it presents a way to explore structures 'membrane-beams' that have a dimensionality between 1D and 2D. 

The observations are well described. The explanation and understanding of the observations could be improved. Some comments and questions:

  1. The motivation of the work might be strengthened, what are the main advantages of this structure/fabrication method (air damping lower, larger surfact to volume ratio for sensing, tuning of resonance frequency by mass/stiffness, controlling nonlinearity?).
  2. Observations are sometimes a bit puzzling (like why is there spring softening)?
  3. Can the structure really be called fractal? I would say that a fractal shows many orders of sizes and self similarity. This device only shows two order of scale, or can any membrane with holes be called fractal?
  4. What are the roles of bending rigidity and tension on the resonance frequencies and modes?
  5. How did you determine the thickness of the suspended silicon and its uniformity? How can thickness be controlled?
  6. For electrostatic actuation the force is proportional to (Voff+Vpp/2 sin wt)^2, so the force terms at frequency w are proportional to Voff*Vpp. When you set Vdc=0 the force is only dc or at frequency 2*w. Please explain why you don't always use Voff nonzero and Voff>>Vpp.

In general I find this an interesting work and recommend publication if the authors address the points mentioned. 

Author Response

1. The main advantage of the structure is its self-similar (fractal) geometry. As it is fabricated, our fractal design provides to the structure the ability of resonating at many different frequencies while keeping quality factors high enough for detection purposes. Our fractal shape also ensures very compact design and larger span to volume ratio.
From the fabrication side, we want to emphasise that our versatile technique successfully fabricate functional devices with complex design which are ready for further investigation. Clearly, we want to show that using this technique one can experiment with sophisticate geometries.
-------------
2. Precise answer for a single spring softening presented in the paper would take an extensive analytical study and experimental efforts. Thus, we have just shown that nonlinearities are present and evenmore all sorts of nonlinearities can be achieved at different frequencies.
Generally, we can approximate a non-linear response of a resonance mode by the Duffing equation (as expalined in reference [44]).
m(d2x/dt2) + c(dx/dt) + k0x + k1x2 + k2x3 = Fcos(ωt) where x is the modal displacement, m is the effective mass, c is the linear damping coefficient, k0 is the linear spring constant, k1 and k2 represent the nonlinear spring constants, and F is the magnitude of the external periodic driving. The modal natural frequency is ω0=sqrt(k0/m) and linear quality factor is Q =ω0m/c. For capacitive micro and nanoresonators, the spring constants include contributions from both mechanical (km) and electrical (ke) restoring forces. Having this, it can be shown that for nonzero k0, the resonant frequency (f) changes as a function of the resonance amplitude (a),
f~f0+ka2 where k=(3k2/8k0-5k12/12k02)f0 (Landau & Lifshitz, Mechanics 1976). Now if k is negaitve or 3k2/8k0-5k12/12k02<0 then we have spring softening.
Also, in reference [45] in the "Supplementary Note 3" a very clear estimation with and explanation of the electrostatic spring softening is given.
It shows that voltages Vac and Vdc are involved in the estimation of k2, namely the term of the cubic spring constant. The voltages Vac and Vdc appear from the harmonic forcing when a parallel plate capacitor model is taken into account and approximation by using Taylor series is applied.

If a nonlinear approximation is applied to every nonlinear peak we possibly might be able to extract Young modulus but we did this in a different manner by using FEM model. Thus, we do not deal analytically with the nonlinearities. We have mentioned that one can explain spring hardening and softening phenomena with the initial stress and the excitation voltage as shown in [44,45] and the references therein. Also, as doping affects both the mechanical and electrical properties of the material (km and ke) it can cause both spring hardening and softening phenomena; as shown in [44].
------------

3. We have seven identical substructures with a diameter of approximately 3.5um. These substructures form a bigger structure that is also a (6,2)-star polygon which gives us the right to define our structure as a (6,2)-star polygon IFS-fractal of second iteration. Still there isn't a precise definition of fractal in the literature but not any membrane with holes could be a fractal. A fractal must have some sort of self-similarity feature. It could be an exact geometrical pattern or a statistically estimated feature that repeats when the object scales. In our case we have defined exactly what is the geometrical shape we deal with, a (6,2)-star polygon that has second iteration in the sense of Iterated Function System [38]. The shape is produced by an IFS fractal generator where one can define the iteration (order of scale) and in our case it is two. Ideally, a fractal must have infinitely many orders of scale but this is practically impossible to be fabricated, hence we decided to start with second iteration in order to check if possible to be fabricated and functionalised. The initial idea was if successful to try with third iteration and compare.

------------

4. By using the FEM software COMSOL, we apply negative tension (compression) to the suspended structure which results in bending.
This way we simulate the effect of residual stress after the fabrication of the device. When we compare the resonance frequencies of the device before and after compression, we saw that when compressed enough, the order of appearance of the different modes is altered.
Without compression we have: even-(0,1), odd-(1,1), even-(1,1), even-(2,1), odd-(2,1)
and after compression we have: even-(1,1), even-(0,1), odd-(1,1), odd-(2,1), even-(2,1)

This result can be seen in the manuscript when Tables 1 and 2 are compared.
-------------
5. The devices have been fabricated by ion implantation of Ga+ at 30 keV. According to SRIM (The Stopping and Range of Ions in Matter) simulations, the implanted and damaged volume in silicon goes up to 40 nm in depth when ions goes perpendicular to the surface (ref 37 in the manuscript). We made measurements by TEM after the ion implantation (ref 35 in the manuscript) and by SEM after the silicon etching (http://dx.doi.org/10.1116/1.4967930 ) and we arrive to this conclusion. Working at the above mentioned conditions the thickness of the devices is fixed to 40 nm.
-------------

6. Initially, we have swiped up and down every peak we detect with Voff=0 and Vpp=10 and Vpp=18 which is as without bias voltage Vdc=0. If an interesting peak appears we applied many different Vpp voltages up to Vpp=20 which is the limit of our instrument. However, for higher frequencies above 20MHz we needed to use Voff in order to obtain better response. Then, we realise that Voff influences the peaks differently than Vpp, hence we did more measurements with different Voff conditions. One such observation was that many double peaks widen when Voff is increased, or some peaks decreased when Voff is increased while others get bigger. These are experimental findings and we do not want to conclude at this stage why theoretically such phenomena appear. However, we find them interesting and we want to share them with the readers.

Reviewer 2 Report

Overall this manuscript is well-written. The reported research is fairly interesting and the presentation is very clear generally.

Comments and questions:

There are some minor grammar issues. for example:

Line 52: our device consists of

Line 85: which --> whose

Line 117: an in-house

How thick is the insulator of SOI wafer? What ions were implanted?   Was thermal annealing carried out after ion implantation and before TMAH etching?

In section 3, it is argued that by increasing the permitivity, charge get distributed further away from the excitation pad of the device. However, what really matters sems to be the reduction of resistivity. So please double check this argument.

Where was the laser focused on the fractal structure?

For figures in section 3, some curves are labeled with “down”. Were other non-labeled curves measured by increasing the frequency?

It is stated that the fractal geometry has a high boundary to volume ratio. However, in some sense, the fractal geometry is made of multiple nanobeams. So how is the boundary to volume ratio higher? Please clarify

Author Response

1.
a) The thickness of the silicon device layer is 2um and the thickness of the SiO2 box is 2um as well.
b) The implanted ions are gallium.
c) After the implantation and the TMAH etching we performed an annealing up to 1000ºC with boron in a nitrogen reach atmotmosphere to promote the recrystallization and doping (p-type) of the device (ref. 35 in the manuscript).
------------

2.

For the clarity of the study we agree that the permitivity could be omitted. This way we have to exclude the importance of the capacitance of the doped material for the overall electrostatic force. If we want to include this effect, it will lead to very technical study and we have to agree on the concept that the excitation force is -(dC/dx)V2/2, where the capacitance is only dependent of the permitivity of the vacuum gap. Thus, the reduced resistivity/impedance increases the Voltage distributed along the fractal structure which leads to increased excitation force.

However, if one wants to study the structure in detail the property of permitivity of the doped material must be included. In reference [40] an experimental study is conducted where p- and n-doped Si sample is characterised for its resistance and capacitance. It is clear from the results that more doping leads to decreased resistance and increased capacitance (increased permitivity) of the doped material. As our resonator is excited by the capacitive force between the electrical contacts to the lower Si-layer and the doped Si-resonator, we need to compute if the electrostatic force will increase when the permitivity of the resonator material increases. There are four layers involved, the fractal structure, the vacuum gap, the insulating SiO2 layer and the bottom Si-layer. By the doping, we change the material properties of the fractal structure and we want to increase the excitation force that is -(dC/dx)V2/2. Hence we want to compute the derivative of the capacitance and to see if it will increase when the permitivity of the top layer increases. If one computes this for four-layer capacitor it indeed increases when the permitivity of the top layer increases. However, an important question arises, if the two material properties resistivity and permitivity were related?
Here I can add few more results from Permitivity and Measurements, V. Komarov, S. Wang, J. Tang, Encyclopedia of RF and Microwave Engineering, 2005 John Wiley & Sons, Inc. If the material is exposed to alternating electric field in RF theory there are different models and one of them suggests complex relative permitivity ε* expressed as ε*=ε'+jε'' where ε' is the dielectric constant representing the stored energy when the material is exposed to an electric field; j=sqrt(1); ε'' is the dielectric loss factor which is the imaginary part, influences energy absorption and attenuation. Further on, ε" can be expressed as ε"=ε"d+ε"σ, where ε"σ=σ/ε0ω; σ-conductivity, ε0-permitivity in vaccum, ω - angular frequency, ε"d is the contribution due to dipole rotation. But let us neglect ε"d, then ε"=σ/ε0ω and the phase angle is expressed as tan(δ)=ε"/ε'=σ/ε0ωε' that relates σ and ε' for the particular frequency of interest. Electric conduction and various polarization mechanisms (including dipole, electronic, ionic) contribute to the dielectric loss factor. An example of using the phase angle of applied alternating current for obtaining experimental resistivity and relative permitivity values of material samples can be found in the study of G. Keller and P. Licastao, Dielectric Constant and Electrical Resistivity of Natural-State Cores, Experimantal and theoretical geophysics, Geological Survey Bulletin, 1052-H.
------------

3. The laser spot of about 12um just covers the whole structure. Therefore, we have aligned the centre of the laser spot with the centre of the structure in order to equally cover the membrane.
------------

4. Yes, the non-labeled curves are measured when the frequency has been swiped up.
------------

5. This is a very good point, the statement has to be rephrased, my suggestion is "high span to volume ratio". Let us consider a circular membrane of radius R, then the surface area is πR2. Now, if we have our truss-structure inscribed in a circle of radius R, where the beam width is R/50 (R=6um, width=0.12um), the surface area would be approximately 7sqrt(3).R2/25. This way we have a structure that is spanning the same area (the area between the pillars) as the circular membrane but when we compare the volumes of our structure and the circular membrane the ratio is 7sqrt(3)/25π=0.154. Now, if we compare the area to the volume of the circular membrane it is 1/h, where h=0.04um is the thickness of the membrane. Let us now consider that the span is the area in between all the pillars or the area of the circular membrane of radius R. Then, the ratio of the the span to the volume of our structure is 25π/(7sqrt(3).h) which is 6.5 times higher than 1/h.